# How to Make Anticancer Drugs Cross the Blood–Brain Barrier to Treat Brain Metastases

**DOI:** 10.3390/ijms21010022

**Published:** 2019-12-18

**Authors:** Eurydice Angeli, Thuy T. Nguyen, Anne Janin, Guilhem Bousquet

**Affiliations:** 1Institut National de la Santé Et de la Recherche Médicale (INSERM), U942, 9 Rue de Chablis, 93000 Bobigny, France; dr.thuynguyen2401@gmail.com (T.T.N.); anne_janin@yahoo.com (A.J.); 2Assistance Publique Hôpitaux de Paris, Avicenne Hospital, Department of medical oncology, 93000 Bobigny, France; 3Medical Oncology Department A, National Cancer Hospital, Ha Noi 110000, Viet Nam; 4AP-HP Saint-Louis Hospital, Laboratory of Pathology, 75010 Paris, France; 5Paris Diderot University/ Université Sorbonne Paris Cité, 5 rue Thomas Mann, 75013 Paris, France; 6INSERM, U1165, 1 Avenue Claude Vellefaux, 75010 Paris, France; 7Paris 13 University, 99 Avenue Jean Baptiste Clément, 93430 Villetaneuse, France

**Keywords:** brain metastases, blood–brain barrier, blood–tumor barrier, copy number profiling, mutation, anticancer drugs, pharmacokinetics

## Abstract

The incidence of brain metastases has increased in the last 10 years. However, the survival of patients with brain metastases remains poor and challenging in daily practice in medical oncology. One of the mechanisms suggested for the persistence of a high incidence of brain metastases is the failure to cross the blood–brain barrier of most chemotherapeutic agents, including the more recent targeted therapies. Therefore, new pharmacological approaches are needed to optimize the efficacy of anticancer drug protocols. In this article, we present recent findings in molecular data on brain metastases. We then discuss published data from pharmacological studies on the crossing of the blood–brain barrier by anticancer agents. We go on to discuss future developments to facilitate drug penetration across the blood–brain barrier for the treatment of brain metastases among cancer patients, using physical methods or physiological transporters.

## 1. Introduction

The incidence of brain metastases in cancer has increased over time in Western countries [1,2], as a result of better control of extra-central nervous system localizations. Melanoma, breast, and lung cancers are the main sources of brain metastases. Following recent improvements in genomic sequencing, targetable genetic alterations are being increasingly identified, with efficient treatments and improvements in survival for most cancers at metastatic stages. However, most pharmaceutical anticancer drugs do not cross the blood–brain barrier (BBB), and many metastatic patients die from brain metastasis as a result, with a short survival rate of less than one year [3]. Indeed, standard care for brain metastases is still surgery or radiation therapy, with limited indications [4,5].

Like the eye, the testis, or the uterus, the central nervous system is a sanctuary site with immune privilege, and it is protected from injury by two mechanisms: a physical restriction barrier, the BBB, and a mechanism involving regulatory immune processes [6].

The BBB is a physiological barrier between the blood and the central nervous system [7], with two major functions: (i) it limits the accumulation of deleterious molecules in the brain by preventing their passage from the blood to the brain, and by favoring their efflux from the brain to the blood. (ii) It maintains brain homeostasis by facilitating active transport of vital endogenous molecules and nutrients. In the neurovascular unit of rat BBB, 10%–15% of all proteins are transporters [8].

Even in case of effraction of the BBB by brain metastasis, the permeability to anticancer drugs remains low. In this review, we discuss the most recent literature to facilitate the crossing of the BBB by anticancer drugs.

## 2. Molecular Biology of Brain Metastasis and Potential Targeted Therapies

Recent technological advances have greatly improved the massive sequencing analysis of tissues from primary cancers and metastases. Parallel sequencing analyses of primary tumors and metastases have been performed in various studies [9,10,11]. Although a primary tumor and its metastases harbor a common signature, some discrepancies have been identified across matched tumor samples from a given patient.

Indeed, since cancers are heterogeneous at cell and molecular levels, metastases can derive from a minority clone in the primary tumor [12]. The metastatic process may result in a clonal selection that shares a common ancestor and yet continues to evolve independently, subject to selective pressure [12,13].

The identification of additional genomic alterations in metastases requires access to metastatic samples. This has been facilitated by the use of imaging-guided biopsies, except in the case of brain metastasis, for which access is restricted to surgical removal or stereotaxic biopsies of single lesions.

There is little data available. For a systematic literature search of comparative studies on genomic profiling between primary and brain metastases, we used the following research algorithm: “Sequence Analysis, RNA” [Mesh] OR “Sequence Analysis, DNA” [Mesh] OR “Molecular Sequence Data” [Mesh] OR “High-Throughput Nucleotide Sequencing” [Mesh] OR “DNA Fingerprinting” [Mesh] OR “DNA Copy Number Variations” [Mesh] OR “DNA Mutational Analysis” [Mesh] AND “Brain Neoplasms/secondary” [Mesh]. A total of 113 articles were initially identified. We then tested a second algorithm: “profiling” [All Fields] AND “brain metastases” [All Fields] AND (“breast cancer” [All Fields] OR “lung cancer” [All Fields] OR “melanoma” [All Fields]). We obtained 69 publications. After selection of recent work (by application of the “5-year” filter), we selected clinical articles on breast cancer, lung cancer, and melanoma. Finally, we retrieved 17 publications (Table 1). While primary tumors and brain metastases usually have a common genomic signature, additional alterations are frequently identified in brain metastases. Interestingly, a comparative study on 14 primary breast cancers and brain metastases demonstrated that 92% of the samples harbored at least one actionable genomic alteration in the brain metastases that was not found in the primary tumor [14]. In addition, it seems that new PIK3CA gene mutations are frequently identified in brain metastases from breast, lung, and melanoma cancers. This suggests that PIK3CA may be implicated in the metastatic process in the brain and should be a preferential target for the development of targeted drugs. In all cases, to improve efficient targeting of brain metastases, a way of overcoming the limitation resulting from the impermeability of the BBB to drugs needs to be sought.

## 3. Is the BBB Disrupted in the Case of Brain Metastases?

The normal BBB (Figure 1a) is composed of endothelial cells with tight junctions, surrounded by a layer of pericytes. The end-feet of the astrocytes wrap around the capillaries and play a role in the formation and maintenance of the BBB [31]. The tight junctions are made up of transmembrane proteins: claudins and occludins anchored in the cytoplasm of the endothelial cells and linked to the cytoskeleton by zona occludens proteins (ZO). Tight junctions limit the paracellular transport of most molecules and pharmacological compounds across the BBB [32,33,34]. A basement membrane, composed of fibronectin, laminin, and collagen, surrounds the endothelial cells and the pericytes [7]. Both endothelial cells and pericytes contribute to the production of the basement membrane [33].

In murine models of brain metastases, there is evidence of disruption and physical changes in the BBB, usually called the blood–tumor barrier (BTB) (Figure 1b). Typically, albumin-bound Evans Blue dye injected intravenously accumulates in brain metastases, while it does not cross the BBB in the absence of metastases [35]. Morphologically, the blood–tumor barrier is anarchic, disorganized, sinuous, irregularly shaped, with large and leaky blood vessels [36]. Ultrathin-section electron microscopy has shown a redistribution of tight junctions on both sides (from the protoplasmic to the exoplasmic side) of the lipid bilayer [37]. The number of normal astrocytes is reduced [38], and they probably lose their ability to produce BBB-inducing factors [39]. The extracellular matrix of tumor vessels is expanded, decreasing the coating of endothelial cells with pericytes. These morphological alterations are linked to molecular alterations: (i) endothelial cells exhibit high levels of vascular endothelial growth factor (VEGF), and CD31 expression (a marker of activation of angiogenesis in endothelial cells) increases [40]. In contrast, tight junction proteins (claudin-1, claudin-5, and 55 kDa occludins) have decreased expression [37,39]. In in vitro models, high VEGF levels are responsible for an increase in the permeability of the BBB by inhibition of tight junction protein expression through the activation of the MAP kinase signaling pathway [41]. (ii) Preclinical models of brain metastases have demonstrated an imbalance in pericyte subpopulations, with an increase in desmin+ and a reduction in CD13+ pericytes, proportionally linked to BBB permeability [40]. (iii) BBB astrocytes have an increased, diffuse expression of aquaporin 4, a water channel. These adaptive modifications may limit the extent of edema fluid intrusion from the tumor bed into the brain extracellular space [38]. (iv) The basement membrane composition is altered, with a decrease in laminin alpha 2, which is correlated with an increase in BBB permeability [40]. In a preclinical model, another study highlighted an upregulation of astrocytic sphingosine-1 phosphate receptor 3 (S1P3) gene expression in brain metastases, responsible for an increase in BTB permeability. The authors demonstrated possibilities of modulation of BTB permeability by S1P3 inhibition or activation [42].

## 4. Limited Brain Delivery of Anticancer Drugs

Several physicochemical parameters are involved in the ability of drugs to cross the BBB and then to remain in the brain: small molecules, liposolubility, charge, interactions with plasma proteins, and interactions with efflux pumps and transporters [36].

For the literature search on the crossing of the BBB or the BTB by drugs, we applied the following method:

We first tested four search algorithms:-(« Blood-brain barrier » [Mesh]) AND (« neoplasm » [Mesh] OR « Cancer » OR « malignant tumor ») AND (« Brain neoplasm » OR « Brain metastasis ») AND (« antineoplasic agents» [Mesh] OR « Chemotherapy » OR « Anticancer drug ») AND (« passage »),-(« antineoplasic agents» [MeSH Terms]) AND (“Blood–Brain Barrier/drug effects” [Majr]) AND (“pharmacokinetic”)-(“anticancer drugs”) AND (“blood brain barrier” (AND (“penetration”)-(“Cerebro Spinal Fluid penetration” (AND (“anticancer drugs” OR “antineoplasic agents”).

We did not identify any relevant papers. We thus extended our literature search using the free text term “cerebro-spinal fluid penetration of antineoplasic agents”. A total of 237 articles were then identified, including 24 articles with pharmacological data (references [43,44,45,46,47,48,49,50,51,52,53,54,55,56,57,58,59,60,61,62,63,64,65]). Overall, drug accumulation in the brain was low, due to limited penetration into the brain and a rapid efflux from the brain to the blood (Table 2). For most targeted therapies, including monoclonal antibodies and tyrosine kinase inhibitors, penetration into brain was barely detectable.

In conclusion, despite morphological and molecular alterations of the BTB, there is almost no pharmacologically relevant penetration by most anticancer drugs.

## 5. Enhancing the Therapeutic Delivery of Drugs from Blood to Brain

### 5.1. Mechanical Disruption of the Blood–Brain Barrier to Deliver Drugs to the Brain

This consists of exerting pressure variations in order to change the conformation of endothelial cells, and permeabilize tight junctions. Two methods have been developed: osmotic disruption and ultrasound disruption.

Osmotic alteration of the BBB, also called osmotic disruption, is known to increase the intracranial and intratumoral concentrations of chemotherapeutic agents [66,67]. In 1973, using electron microscopy, Brightman et al. visualized a disruption of BBB tight junctions in rats after intracarotid perfusion of a hyperosmotic solution of urea [68]. A hyperosmotic solution produces a transient, reversible disruption of the BBB by causing endothelial cell shrinkage and thus opening the tight junctions (Figure 2A). In clinical practice, the procedure includes an intra-arterial infusion of hypertonic solution of mannitol (25%) into a carotid or vertebral artery, followed by intra-arterial delivery of chemotherapy, for the treatment of brain tumors. Knuutinen et al. [69] recently reported the case of a 25-year-old man with brain metastases of a germ cell tumor, progressing after standard treatments. After two months of osmotic BBB disruption with an intra-arterial association of carboplatin, cyclophosphamide, and etoposide, the patient achieved a complete, durable response. Comparable promising results have been reported with brain metastases or primary tumors from other cancer types [67,70]. Apart from the clinical benefit, this method is also safe, with toxicity profiles similar to intravenous chemotherapy. However, it is difficult to implement this procedure in daily practice since it requires several days of hospitalization with intra-arterial cranial catheter placement under general anesthesia [71].

The use of extracranial ultrasounds has been explored to improve the delivery of drugs to brain tumors. This technique is combined with intravenous administration of air microbubbles and has the advantage of being noninvasive. When excited by ultrasounds, microbubbles expand and exert a mechanical force on the endothelial cells of the BBB, leading to tight junction disruption [72] and also to increased activity of active transports [73] (Figure 2B). Preclinical studies using ultrasound disruption for the delivery of anticancer drugs have recently been reviewed [74]. The use of focused ultrasounds can lead to an average 4-fold increase in the delivery of chemotherapeutic agents and a 3.5-fold increase in the delivery of monoclonal antibodies [74].

Ultrasound delivery can also be radiologically guided by magnetic resonance imagery (MRI). One early clinical study has demonstrated the safety of using ultrasounds combined with microbubbles to disrupt the BBB, in combination with carboplatin, in 17 patients with glioblastoma. This pilot study opens new perspectives for optimizing chemotherapy delivery to brain tumors [75].

### 5.2. Intranasal Delivery of Drugs

In 2004, a study demonstrated that after intranasal administration, xenobiotics were able to penetrate the central nervous system through the olfactory and trigeminal pathway [76]. Since then, intranasal drug delivery has been offered for the treatment of brain tumors. Preclinical studies using murine models of brain tumors demonstrated a better cerebral absorption of drugs following intranasal delivery [77,78]. In a nonhuman primate model, the CSF/serum concentration ratio of temozolomide was 22% after intravenous administration, compared to 36% after intranasal delivery [79]. Different preclinical and clinical trials using intranasal administration have been developed and are reviewed by Peterson et al. [80].

The anticancer drug perillyl alcohol, a chemical derived from essential oil, has been selectively developed to treat central nervous system tumors using intranasal delivery. In a phase-II clinical study of 37 patients with recurrent malignant glioma [81], perillyl alcohol intranasal delivery was well tolerated. Furthermore, when compared to historically untreated controls, patients with primary recurrent glioblastoma treated with intranasal perillyl alcohol showed a 5.9-month survival advantage [82]. After 4 years of exclusive perillyl alcohol inhalation treatment of 198 patients with recurrent malignant glioma, 19% of the patients were still in clinical remission [83]. In addition, the toxicity profile of intranasal administration seems to be good compared to systemic administration. A multicenter phase 1/2a trial started in the United States in 2016, including patients with recurrent glioblastoma (ClinicalTrials.gov identifier: NCT02704858). It uses NEO100, a synthetic, highly pure version of perillyl alcohol. Preliminary results are expected in 2020 [84].

### 5.3. The Use of Nanoparticles to Cross the BBB

Nanoparticles are natural or artificial particles ranging from 10 and 1000 nm in size [85]. The term nanoparticle applies to a wide variety of drug delivery vehicles, including dendrimers, micelles, liposomes, nanoscale ceramics, metallic and polymer nanoparticles [86]. Nanoparticles have two advantages: first, they are taken up by the BBB endothelial cells (by receptor-mediated or adsorptive-mediated transcytosis pathways), and secondly, when administered systemically, nanoparticle encapsulation protects the drugs transported from damage. A wide variety of nanoparticles have been developed and are described by Khaitan et al. [87]. Here, we focused on nanoparticles specifically developed to treat brain tumors.

Polymeric nanoparticles are made of synthetic polymers [86]. Different in vitro/in vivo studies showed better cellular uptake, cytotoxic effects, and brain penetration with drugs bound to polymeric nanoparticles [88,89,90]. However, while a better brain uptake is observed when antitumoral drugs are associated with polymeric nanoparticles, brain penetration is mainly due to the combination with polysorbate 80 (a nonionic surfactant and emulsifier), which interacts with the low-density lipoprotein receptor by an endocytic process [85]. After intravenous injection of doxorubicin in normal rats, brain concentrations can be enhanced more than 60-fold when doxorubicin is linked to polymeric nanoparticles coated with polysorbate 80 [91]. In a model of rat glioblastoma, 40% of rats treated with this formulation survived for half a year [92]. Furthermore, the toxicity profile was similar or even better than that of free doxorubicin.

Gold nanoparticles have recently emerged as innovative tools, particularly for photothermal therapy since they can be excited by laser to induce hyperthermia and cytotoxicity in tumor cells. Gold nanoparticles have already been tested for both diagnosis and therapy, and they present a low toxicity profile. Gold nanoparticles can also be functionalized to target cancer cells. Our team recently demonstrated a significant inhibition of tumor growth in a mouse xenograft model of HER2-overexpressing breast cancer, using functionalized gold nanoparticles with anti-HER-2 antibodies [93]. For brain tumors, gold nanoparticles seem to be of particular interest. In an in vitro blood–brain barrier model, glucose-coated gold nanoparticles are taken up by brain-endothelium, and they target glial cells by passive diffusion. Furthermore, gold nanoparticles are able to selectively penetrate the brain endothelial cells, rather than other endothelial cells [94] (Figure 2C). Jensen et al. [95] developed gold nanoparticles conjugated with nucleic acids targeting the oncoprotein Bcl2Like12 in glioblastoma xenografts. They demonstrated an efficient penetration of this nanoparticle across the BBB, with a significant antitumor effect [95]. A clinical trial using this nanoparticle (called NU-0129) is ongoing for the treatment of recurrent glioblastoma.

Liposomes are vesicles made up of a phospholipid bilayer and an aqueous core. Vesicle size is typically between 50 nm and 5 μm. They have several advantages, such as the possibility of carrying various types of drugs. They can pass through the inter-endothelial gaps in the leaky vasculature of the BTB, or by active receptor-mediated transports when they are linked to specific ligands [96] (Figure 2C). In preclinical studies, they tend to accumulate in brain tumor tissues rather than in normal brain tissues [97], thus favoring brain penetration by drugs. In a rat model, brain penetration of docetaxel loaded on liposomes is greatly enhanced (100%) compared to docetaxel alone [98]. In a normal model of breast cancer brain metastasis, irinotecan was compared to a liposomal formulation (nal-IRI-50): the irinotecan metabolite SN38 accumulated in the brain metastases at 7 days after intravenous injection of nal-IRI-50, while it was undetectable with irinotecan 12 h post-administration [99].

In 15 patients with glioblastoma or metastatic brain tumors, radiolabeled liposomal doxorubicin strongly accumulated in brain tumors with significant responses in 14 patients [97]. The same liposomal formulation of doxorubicin functionalized with anti-HER2 antibodies (^64^Cu-MM-302) also accumulated in HER2 overexpressing breast cancer brain metastases [100].

### 5.4. Combinations of Drugs Targeting Receptor-Mediated Transport

Receptor-mediated transport, expressed in the endothelial cells of the BBB, enables active transporters to cross the BBB to maintain homeostasis of the central nervous system. Typically, the insulin receptor is responsible for the import of insulin from blood to brain. The low-density lipoprotein receptor mediates the transport of lipoproteins and of many other ligands across the BBB. The transferrin receptor mediates iron transport [101]. When a ligand binds to its specific receptor on endothelial cells of the BBB, it leads to the endocytosis of the ligand-receptor complex into a transport vesicle. When the vesicle reaches the opposite membrane, it fuses with the membrane and enables the ligand to be released into the brain compartment (Figure 2C). Using these physiological shuttles, bispecific antibodies or drug receptor conjugates have been developed.

Transferrin receptor is of particular interest, since it is overexpressed in cancers. Since 1981 and the first anti-transferrin receptor antibody conjugate, several compounds have been investigated (reviewed in [102]). Tf-CRM107, a conjugate of human transferrin and diphtheria toxin, has been tested after intraventricular injection in a phase-II study on 44 patients with glioblastoma or astrocytoma [103,104]. A response was obtained in 12 patients. Unfortunately, the phase-III clinical trial failed to demonstrate a benefit due to unacceptable toxicities. More recently, drug conjugates targeting transferrin receptor in nanoparticle formulations are being tested by intravenous administration, with promising pharmacological and antitumor results in preclinical studies [105,106,107].

One limitation remains, linked to toxicities due to the ubiquitous expression of transferrin receptor in humans.

Lipoproteins are responsible for the blood transport of hydrophobic molecules including triglycerides and cholesterol. Apolipoproteins are parts of lipoproteins, ensuring cohesion and solubilization of lipoproteins in the blood. Depending on the different types of apolipoproteins present on their surface, the lipoproteins are captured by lipoprotein receptors of specific cells. Drug nanoparticles conjugated with apolipoprotein domains have been developed with limited brain penetration due to their large size and their competition with circulating endogenous ligands [108,109].

Other molecules contain a Kunitz domain, a region of 50–60 specific amino acids, which is the ligand of lipoprotein receptors. This led to the identification of a family of peptides, named angiopeps, derived from the Kunitz domain, with good transcytosis ability from blood to brain. In 2008, Regina et al. developed a new drug combining paclitaxel and angiopep2, called ANG1005 or GRN1005, with promising preclinical results in glioma or brain metastases [110]. A phase-1 clinical trial has been successfully completed on brain metastases in different cancers, and in recurrent malignant gliomas [111,112]. In a phase-II study on women with breast cancer brain metastasis or leptomeningeal carcinomatosis, the clinical benefit was 71%, with a median survival of 8 months [113]. A Phase 3-trial is underway with results expected in 2020.

This promising data led to the engineering of Angiopep2 conjugates with targeted therapies: a combination of an anti-epidermal growth factor reception (EGFR) and Angiopep2, with promising results in murine models of glioblastoma [114], and a combination of Angiopep2 with a new anti-HER2 antibody, with a strong brain uptake in xenograft models of human epidermal receptor (HER)-overexpressing breast cancer brain metastasis [115].

## 6. Conclusions

The treatment of brain metastases remains a major therapeutic challenge since standard treatments (surgery, radiation therapy) are of limited benefit. Innovative approaches using physical methods or physiological transporters are being explored to facilitate drug penetration across the blood–brain barrier and deliver them to brain metastases at relevant pharmacological concentrations.

Drug conjugates using mediate transport receptors, particularly angiopeps, are the most promising therapies with ongoing phase III clinical studies. They open avenues for further clinical applications to facilitate the passage of targeted therapies across the BBB and thus target molecular abnormalities linked to brain tumors.

## Figures and Tables

**Figure 1 ijms-21-00022-f001:**
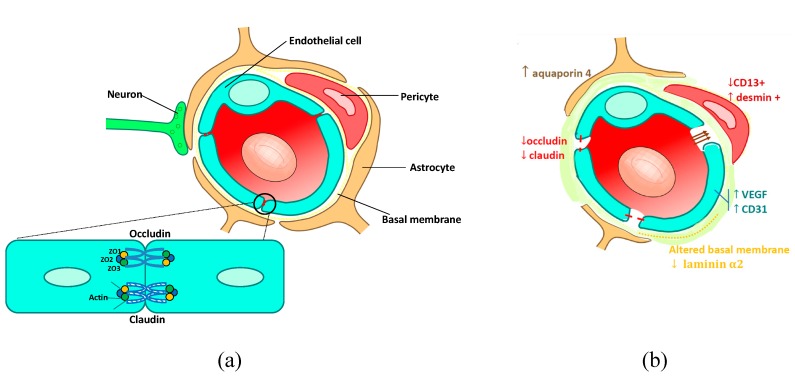
Neurovascular unit of (**a**) normal blood–brain barrier (BBB) and (**b**) blood–tumor barrier (Credits images: © User: Kuebi / Wikimedia Commons / CC-BY-3.0), ↑ means an increase,↓ means a decrease.

**Figure 2 ijms-21-00022-f002:**
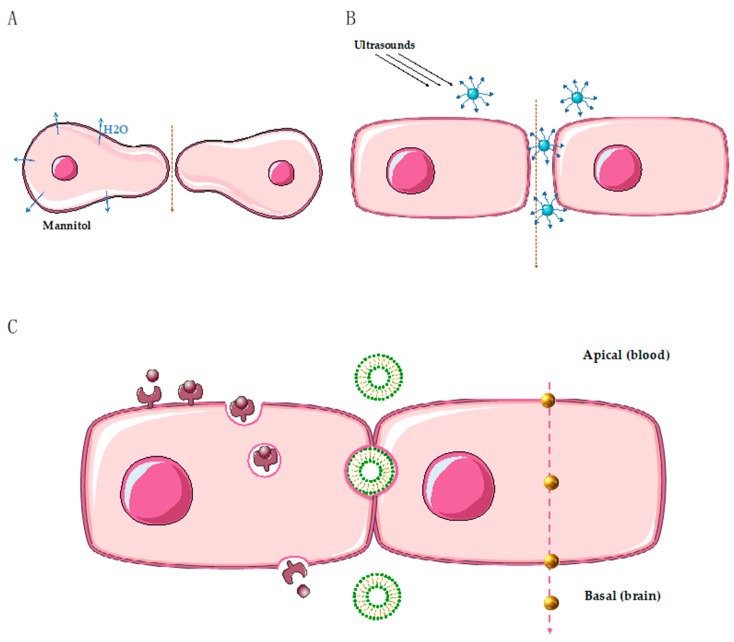
Summary of different way to overcome the BBB (except the intranasal). (**A**) Osmotic disruption, hypertonic mannitol causes a water leakage to the extracellular area and a shrinkage of endothelial cells (blue arrow means an extravasation of H2O from intracellular to extracellular space/ pink arrow means the passage of drugs across the BBB from blood to brain). (**B**) Ultrasounds combined to microbubbles: when excited by ultrasounds, microbubbles expand and exert a mechanical force on the endothelial cells of the BBB, leading to tight junction disruption. (**C**) Transcytosis across endothelial cells of the blood–brain barrier. Left side: receptor mediated transport: binding of the ligand to a specific receptor on the apical side, invagination of the membrane containing the complex, transcytosis, fusion, and release of the cargo to the basal side. Middle: inter-endothelial passage of liposomal nanoparticles. Right side: passive diffusion of gold nanoparticles (Credits images: © SMART / CC-BY-3.0).

**Table 1 ijms-21-00022-t001:** Molecular alterations in brain metastases.

Primary Localization	Histological Subtype	Patient Numbers/ BM Analyzed	Materials and Analyses	Examples of Driver mutations or CNV Acquired in BM	Reference
**Breast**	All	14/14	DNA	PIK3CA EGFR FGFR1	Tyran 2019 [14]
	All	45/42	DNA	TP53 (ns)	Lee JY 2015 [15]
	All	61/61	RNA Protein	SOX2 OLIG2 ERBB2	Lee JY 2016 [16]
	All	21/21	RNA Protein	RET HER2 amplification	Varešlija 2019 [17]
	ER negative	17/ 9	DNA	TP53 PIK3CA SMAD4 RB1	Schrijver 2018 [18]
	All	78/52	DNA RNA protein	PIK3CA HER3 EGFR HRAS/KRAS/NRAS	Da Silva 2010 [19]
	All	10/10	DNA Protein	PTEN FBXW7 ERBB2 KIT	Bollig-Fisher 2015 [20]
	All	35/3	DNA RNA Protein	TP53 ERBB2 BRCA1–2 IDH1 CDH1	Schulten 2017 [21]
	All	20/20	DNA RNA Protein	ERBB2 FGFR4 EGFR ESR1	Priedigkeit 2017 [22]
**Lung**	Squamous	79/9	DNA RNA Protein	Total of 23 genes	Paik 2015 [23]
	NSCLC	61/61	DNA	PIK3CA EGFR MET ROS1 VEGFA CCND1 CDKN2A/2B	Wang 2019 [24]
	NSCLC	1/1	DNA Protein	PTEN CDKN2A	Li 2015 [25]
**Melanoma**		16/16	DNA RNA Protein	PI3K/AKT pathway genes	Chen G 2014 [26]
		74/88	DNA RNA Protein	Increase in oxidative phosphorylation gene expression	Fisher 2019 [27]
**Both**	Lung Breast Melanoma	NA/493	DNA protein	TOP2A cMET (melanoma) HER2 (breast) EGFR	Ferguson 2018 [28]
	Lung Breast Renal carcinoma	86/86	DNA	PTEN EGFR PI3K/AKT pathway genes HER2 amplification MCL1 amplification	Brastianos 2015 [29]
	Breast Lung Melanoma Esophagus	36/36	DNA RNA	MAP3K4 COL5A1	Saunus 2015 [30]

BM: brain metastases; CNV: Copy Number Variations; NSCLC: Non-Squamous Cell Lung Cancer; ns: nonsignificant.

**Table 2 ijms-21-00022-t002:** Cerebrospinal fluid (CSF) pharmacokinetics for anticancer drugs.

Drug Family	CSF/Plasma Ratio (%)	Species Studied	Time	Ref
**Chemotherapy**				
Thiotepa	100 (for thiotepa and metabolite)	Human (children)	AUC _0–24 h_	[43]
Temozolomide	20	Human	AUC _0–5 h_	[44]
Methotrexate	2.8	Human (children)	24 h	[45]
Topotecan	32	RHM	AUC _0–60 min_	[46]
Irinotecan	9.6–16 Metabolite SN 38: <3	RHM	AUC _0–48 h_	[47]
Cisplatin	3	RHM	AUC _0–4 h_	[48]
Carboplatin	2.6	RHM	AUC _0–4 h_	[48]
Oxaliplatin	1.2	RHM	AUC _0–4 h_	[48]
Etoposide	0–3	Human (children)	Mean _0–5 h_	[49]
Doxorubicin	<5	RHM	Mean _0–48 h_	[50]
Idarubicin	0–15 Metabolite idarubicinol: 1.9	RHM	1 h	[51]
Daunorubicin	2.4 (Metabolite Daunorubicinol)	RHM	AUC _0–96 h_	[51]
Tomudex	0.6–2.0	RHM	Mean _0–48 h_	[52]
Docetaxel	0.1–9	Human	72 h	[53]
Pemetrexed	1–3	Human	AUC _1–4 h_	[54]
	0.76	RHM	AUC _0–__∞_	[55]
Ciclofosphamid	17	RHM	AUC _0–240 min_	[56]
Ifosphamid	38	RHM	AUC _0–240 min_	[57]
	Metabolite 4-OH-Ifo: 30			
	13	RHM	AUC _0–240 min_	[56]
Vincristin	0	Human (children)	Mean _8–46 min_	[58]
	0	Human	Mean _0–24 h_	[59]
Gemcitabin	6.7	RHM	NA	[60]
TKI				
Gefitinib	1	Mice	1 h	[61]
Erlotinib	1	Mice	1 h	[61]
Icotinib	0.7	Mice	1 h	[61]
Imatinib	5	RHM	AUC _0–48 h_	[62]
Osimertinib	>100 (brain/plasma ratio)	Mouse and monkey	AUC _0–90 min_	[63]
Antibodies				
Trastuzumab	0.5	Rat	AUC _0–722 h_	[64]
Rituximab	0.2	Human	Mean _0–15 days_	[65]

CSF: cerebrospinal fluid; AUC: area under the curve; RHM: Rhesus Monkey; TKI: Tyrosine kinase inhibitors.

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
