# Peer review of "How to Make Anticancer Drugs Cross the Blood–Brain Barrier to Treat Brain Metastases"

_ijms, 2019, doi:10.3390/ijms21010022_

Round 1

Reviewer 1 Report

Review of “How to make anticancer drugs cross the blood-brain-barrier “

Summary: The authors in this review article discussed issues in the treatment of brain metastases such as, the difficulties in genetically profiling brain metastasis and transporting anti-cancer drugs across the blood brain barrier (BBB) to treat brain metastases. The authors also summarized recent techniques for treatment of brain metastases, with regards to direct targeting of the BBB and various alterations in the way the drugs are delivered.

Overall Strengths/ limitations:

The strengths of the paper are that it is well organized and the methodology in how the literature search was performed was explained in detail.

The major limitation of this review article is the very short conclusion that does not add much to the overall paper. The paper also does not address how the studies described could improve the currently used treatments for brain metastasis. Further, while figure one is beautiful and informative, figure 2 further show a lack of effort like the conclusion and adds little or nothing.

Methods/results

The method used for the literature search were well explained and there are no issues with the presentation of the figures and tables.

Overall this article was well written and provides a concise review of the current state in treatment of brain metastases. Before publication the authors should expand on the findings of this study as well as address how the treatments discussed compares to the current treatments for brain metastases. Also line 168 should be indented.

This paper is very scholarly however, please summarize major areas reviewed in paper... in the very least in the conclusion.  Take this information and suggestion where the future might be for getting drugs across the BBB.

Author Response

Point 1: The strengths of the paper are that it is well organized and the methodology in how the literature search was performed was explained in detail.

Response 1: We thank Reviever#1 for his/her nice comment.

Point 2: The major limitation of this review article is the very short conclusion that does not add much to the overall paper. The paper also does not address how the studies described could improve the currently used treatments for brain metastasis.

Response 2: To follow the advice by Reviewer#1, we have now extended the conclusion of our manuscript, enlightening the therapeutic perspectives to improve survival of patients with brain tumors, including cancer brain metastases. We have now written in the revised version of the manuscript, page 9, lines 293 to 300 “The treatment of brain metastases remains a major therapeutic challenge since standard treatments (surgery, radiation therapy) are of limited benefit. Innovative approaches using physical methods or physiological transporters are being explored to facilitate drug penetration across the blood-brain barrier and deliver them to brain metastases at relevant pharmacological concentrations. Drug conjugates using mediate transport receptors, particularly angiopeps, are the most promising therapies with ongoing phase III clinical studies. They open avenues for further clinical applications to facilitate the passage of targeted therapies across the BBB and thus target molecular abnormalities linked to brain tumors”

Point 3: Further, while figure one is beautiful and informative, figure 2 further show a lack of effort like the conclusion and adds little or nothing.

Response 3: To follow the advice by Reviewer#1, we have thoroughly redrawn Figure 2 which is now illustrating the innovative approaches described in our review to make anticancer drugs cross the blood-brain barrier.

Point 4: The method used for the literature search were well explained and there are no issues with the presentation of the figures and tables. Overall this article was well written and provides a concise review of the current state in treatment of brain metastases. Before publication the authors should expand on the findings of this study as well as address how the treatments discussed compares to the current treatments for brain metastases.

Response 4: We thank Reviewer#1 for his/her comments and advices. We have now rewritten the conclusion in the revised version of the manuscript, page 9, lines 293 to 300 “The treatment of brain metastases remains a major therapeutic challenge since standard treatments (surgery, radiation therapy) are of limited benefit […] Drug conjugates using mediate transport receptors, particularly angiopeps, are the most promising therapies with ongoing phase III clinical studies. They open avenues for further clinical applications to facilitate the passage of targeted therapies across the BBB and thus target molecular abnormalities linked to brain tumors”

Point 5: Also line 168 should be indented.

Response 5: We have now added an indentation to this line (line 169 in the new version)

Point 6: This paper is very scholarly however, please summarize major areas reviewed in paper... in the very least in the conclusion.  Take this information and suggestion where the future might be for getting drugs across the BBB.

Response 6: We thank Reviewer#1 for his/her comments and advices. We have now rewritten the conclusion in the revised version of the manuscript, page 9, line 294 to 296 “Innovative approaches using physical methods or physiological transporters are being explored to facilitate drug penetration across the blood-brain barrier and deliver them to brain metastases at relevant pharmacological concentrations”.

Furthermore, we have thoroughly redrawn Figure 2 that summarizes the different way to overcome the BBB.

Reviewer 2 Report

In this review Angeli et al. have presented a very thorough literature review of how the present concepts can be applied in order to target Blood Brain Barrier (BBB) in brain tumor in order to effectively target them.

They started describing the molecular biology of BBB and how it prevent anything from entering into BBB. Then they discussed the challenges of targeting brain tumor because of BBB. They further discussed strategies that are tried in present and in the past and how it can be improved.

Overall, this review is quite informative and can be accepted in its current form.

Author Response

Point 1: In this review Angeli et al. have presented a very thorough literature review of how the present concepts can be applied in order to target Blood Brain Barrier (BBB) in brain tumor in order to effectively target them. They started describing the molecular biology of BBB and how it prevent anything from entering into BBB. Then they discussed the challenges of targeting brain tumor because of BBB. They further discussed strategies that are tried in present and in the past and how it can be improved.

Overall, this review is quite informative and can be accepted in its current form.

Response 1: We thank Reviewer#2 for this recommendation.

Round 2

Reviewer 1 Report

Great work on these revisions!  Solid paper.